# SARS-CoV-2 Nonstructural Proteins 1 and 13 Suppress Caspase-1 and the NLRP3 Inflammasome Activation

**DOI:** 10.3390/microorganisms9030494

**Published:** 2021-02-26

**Authors:** Na-Eun Kim, Dae-Kyum Kim, Yoon-Jae Song

**Affiliations:** 1Department of Life Science, Gachon University, Seongnam-Si, Gyeonggi-do 13120, Korea; kne951027@naver.com; 2Donnelly Centre, University of Toronto, Toronto, ON M5S 3E1, Canada; daekyum.kim@utoronto.ca; 3Department of Molecular Genetics, University of Toronto, Toronto, ON M5S 3E1, Canada; 4Lunenfeld-Tanenbaum Research Institute, Sinai Health System, Toronto, ON M5G 1X5, Canada; 5Center for Personalized Medicine, Roswell Park Comprehensive Cancer Center, Buffalo, NY 14203, USA

**Keywords:** severe acute respiratory syndrome coronavirus 2, nonstructural protein, inflammasome, caspase-1

## Abstract

Viral infection-induced activation of inflammasome complexes has both positive and negative effects on the host. Proper activation of inflammasome complexes induces down-stream effector mechanisms that inhibit viral replication and promote viral clearance, whereas dysregulated activation has detrimental effects on the host. Coronaviruses, including SARS-CoV and MERS-CoV, encode viroporins that activate the NLRP3 inflammasome, and the severity of coronavirus disease is associated with the inflammasome activation. Although the NLRP3 inflammasome activation is implicated in the pathogenesis of coronaviruses, these viruses must evade inflammasome-mediated antiviral immune responses to establish primary replication. Screening of a complementary DNA (cDNA) library encoding 28 SARS-CoV-2 open reading frames (ORFs) showed that two nonstructural proteins (NSPs), NSP1 and NSP13, inhibited caspase-1-mediated IL-1β activation. NSP1 amino acid residues involved in host translation shutoff and NSP13 domains responsible for helicase activity were associated with caspase-1 inhibition. In THP-1 cells, both NSP1 and NSP13 significantly reduced NLRP3-inflammasome-induced caspase-1 activity and IL-1β secretion. These findings indicate that SARS-CoV-2 NSP1 and NSP13 are potent antagonists of the NLRP3 inflammasome.

## 1. Introduction

Severe acute respiratory syndrome coronavirus 2 (SARS-CoV-2), the etiological agent responsible for coronavirus disease 2019 (COVID-19), belongs to the β-coronavirus genera, which include SARS-CoV and Middle East respiratory syndrome coronavirus (MERS-CoV) [1]. SARS-CoV-2 has a single-stranded positive-sense RNA (+ssRNA) genome, which acts as a template in the direct translation of two overlapping open-reading frames (ORFs), ORF1a and ORF1b, which generate the polyproteins pp1a and pp1ab, respectively. These polyproteins are further processed by two viral proteases, a papain-like protease called NSP3 and a 3C-like protease, NSP5, into 16 NSPs. In addition, subgenomic RNAs generated by the discontinuous transcription of negative-strand RNAs generate ORFs that encode the four major structural proteins of SARS-CoV-2, S, E, M and N, along with various accessory proteins [2].

Upon virus infection, pattern recognition receptors (PRRs) on host cells sense viral pathogen-associated molecular patterns (PAMPs) and activate the assembly of intracellular multiprotein complexes known as inflammasomes [3]. Upon sensing viral proteins and/or nucleic acids, the nucleotide-binding domain (NBD) leucine-rich repeat (LRR)-containing receptor (NLR) family pyrin domain containing 3 (NLRP3) and the pyrin and hematopoietic interferon-inducible nuclear (PYHIN) domain proteins, including absent in melanoma 2 (AIM2) and interferon- γ (IFN-γ) inducible protein 16 (IFI16) form canonical inflammasome complexes by functioning as scaffold proteins to recruit the adaptor proteins apoptosis-associated speck-like protein containing a caspase recruitment domain (CARD) (ASC) and pro-caspase-1 [3]. Oligomerization induces the auto-proteolytic activation of pro-caspase-1 to caspase-1. Caspase-1, in turn, cleaves inactive pro-IL-β and pro-IL-18 to yield active IL-1β and IL-18, which activate subsequent systemic immune responses to limit viral infection and spread [3]. Inflammasome activation also induces a type of inflammatory cell death, known as pyroptosis of infected cells, which releases damage-associated molecular patterns (DAMPs) that further promote inflammatory responses.

To productively infect hosts, viruses must evade host immune responses, including inflammasomes, and reproduce. Viruses have been shown to counteract inflammasome activation by down-regulating the expression of inflammasome components, including NLRP3, ASC and IL-1β, blocking the assembly of inflammasomes and/or inhibiting caspase-1 activity [4,5,6,7,8,9,10,11,12,13,14,15,16,17,18,19,20].

SARS-CoV has been reported to activate NLRP3 inflammasomes [21,22,23,24]. Unregulated inflammasome activation and proinflammatory cytokine expression are associated with coronavirus pathogenesis [25,26,27,28]. Viral proteins, such as E, ORF3a and ORF8b, produced by subgenomic RNAs, have been found to be responsible for NLRP3 inflammasome activation by inducing intracellular ionic imbalances and endoplasmic reticulum (ER) stress. For example, the E protein was shown to form calcium ion channels in ER Golgi apparatus intermediate compartment (ERGIC)/Golgi membranes and activate NLRP3 inflammasomes [22]. ORF3a also activates NLRP3 inflammasomes, possibly by modulating potassium ion efflux and mitochondrial production of reactive oxygen species (ROS) [21,24]. ORF8b induces ER stress and lysosomal damage, which, in turn, activate NLRP3 inflammasomes [23].

Although SARS-CoV-2 must evade inflammasome responses to establish primary replication in infected hosts, the strategies used by viruses to counteract inflammasome activity have not been determined. This study, therefore, attempted to identify SARS-CoV-2 proteins interfering with inflammasome activation using a complementary DNA (cDNA) library containing SARS-CoV-2 ORFs. 

## 2. Results

### 2.1. Screening of SARS-CoV-2 cDNA Libraries to Identify Viral ORFs That Inhibit Caspase-1 Activity

To identify SARS-CoV-2 ORFs that inhibit caspase-1-mediated IL-1β activation, a cDNA library consisting of 28 SARS-CoV-2 ORFs was screened using an enzyme-linked immunosorbent assay (ELISA) specific to mature IL-1β. To establish the assay for caspase-1 activity, 8.0 × 10^5^ human embryonic kidney 293T (HEK293T) cells in each well of a six-well plate were transfected with a vector expressing pro-IL-1β (20 ng), with or without a vector expressing caspase-1 (200 ng), and concentrations of mature IL-1β in culture supernatant were measured using IL-1β ELISA. Expression of pro-IL-1β alone in HEK293T cells did not significantly affect the secretion of mature IL-1β (16.5 pg/mL), whereas the secretion of IL-1β was increased 23.6-fold, to 390 pg/mL, in HEK293T cells co-expressing pro-IL-1β and caspase-1 (Figure 1). 

To determine the effects of SARS-CoV-2 ORFs on caspase-1-mediated IL-1β activation, HEK293T cells were co-transfected with vectors expressing caspase-1 and pro-IL-1β, plus the expression vectors for SARS-CoV-2 ORFs, and IL-1β ELISA was performed as described above. Of these, SARS-CoV-2 ORFs, NSP1 and NSP13 reduced caspase-1-mediated IL-1β secretion by 91% and 71%, respectively, whereas other ORFs did not exhibit significant effects (Figure 1). These findings indicated that NSP1 and NSP13 are potent inhibitors of caspase-1-mediated IL-1β activation.

### 2.2. The Amino Acid Residues of NSP1 Critical for Host Translation Shutoff Are Required for Caspase-1 Inhibition

NSP1 has been shown to bind the 40S ribosome subunit, inhibiting the translation and inducing the degradation of the host’s messenger RNA (mRNA) without affecting viral RNAs [29,30,31,32]. NSP1 containing the mutations R124A/K125A and K164A/H165A, however, is unable to promote RNA cleavage and suppress host translation [33,34]. To determine whether NSP1 suppression of host mRNA translation is critical for caspase-1 inhibition, HEK293T cells were transfected with vectors expressing caspase-1 and pro-IL-1β, along with expression vectors for NSP1 wild-type (WT) and NSP1 carrying the mutations R124A/K125A, K164A/H165A, and both R124A/K125A and K164A/H165A (quadruple mutant (QM)) and deletion of amino acids 160–173 (∆ 160–173). After 24 h, the concentrations of mature IL-1β in culture supernatant were measured by ELISA. Although NSP1 WT strongly abrogated the secretion of mature IL-1β, none of the NSP1 mutants had any effects (Figure 2A).

The effect of NSP1 on the cleavage of caspase-1 into its p20 subunit was tested by western blotting (Figure 2B). As previously reported, levels of NSP1 proteins containing the mutations R124A/K125A, K164A/H165A, QM and ∆ 160–173 were higher than those of NSP1 WT [34]. NSP1 WT significantly reduced the levels of caspase-1 protein and blocked the cleavage of caspase-1 into its p20 subunit. In contrast, none of the NSP1 mutants had any effects on caspase-1 protein levels or its cleavage into p20 subunit. In addition, NSP1 WT, but none of the NSP1 mutants, reduced the levels of caspase-1 and IL-1β mRNAs (Figure 2A). These findings indicate that NSP1 promotion of RNA cleavage and suppression of host translation are critical for caspase-1 inhibition.

### 2.3. Both the N- and C-Termini of NSP13 Protein Are Required for Caspase-1 Inhibition

To determine the domains of NSP13 that are critical for caspase-1 inhibition, HEK293T cells were co-transfected with vectors expressing caspase-1 and pro-IL-1β, along with expression vectors for NSP13 WT and mutants carrying deletions of the zinc-binding domain (ZBD) (∆ZBD; amino acids 1–99), the stalk domain (∆S; amino acids 100–149), the 1B domain (∆1B; amino acids 150–260), the two RecA-like domains, 1A (∆1A; amino acids 261–441), 2A (∆2A; amino acids 442–601) and both 1A and 2A (∆1A/2A; amino acids 261–601) (Figure 3A). After 24 h, the concentrations of mature IL-1β in culture supernatant were measured by ELISA (Figure 3B). Although NSP13 WT significantly reduced the secretion of mature IL-1β, none of the NSP13 mutants had any effects. Similarly, NSP13 WT, but none of the mutants, inhibited the cleavage of caspase-1 into its active p20 subunit without affecting the levels of caspase-1 protein (Figure 3C). Neither NSP13 WT nor any of the mutants had any effects on levels of caspase-1 and IL-1β mRNA (Figure 2B). These results indicate that both the N-terminal ZBD and the C-terminal helicase domains of NSP13 are required for caspase-1 inhibition. 

### 2.4. NSP1 and NSP13 Inhibit Activation of the NLRP3 Inflammasome in THP-1 Cells

To test the effects of NSP1 and NSP13 on NLRP3 inflammasome activation, THP-1 cells were electroporated with a vector expressing NSP1 WT, NSP1 QM, NSP13 WT or NSP13 ∆1A/2A. The electroporation efficiency, which was determined by electroporating these cells with a GFP-expression vector and by calculating the percentage of cells expressing GFP using the Neon™ transfection system, was approximately 20%. Twenty hours after electroporation, the THP-1 cells were differentiated by incubation with phorbol-12-myristate 13-acetate (PMA) and subsequently treated with lipopolysaccharide (LPS) and adenosine triphosphate (ATP) to activate the NLRP3 inflammasome. 

NLRP3-inflammasome-induced caspase-1 activation was determined using the Caspase Glo^®^1 Inflammasome assay (Figure 4A). Specific caspase-1 activity was determined by subtracting caspase activities in the supernatant of cells treated with the caspase-1 inhibitor YVAD-CHO from activities in the supernatants of cells not treated with the inhibitor. Treatment of THP-1 cells with LPS- and ATP-activated caspase-1 52-fold (Figure 4A). Both NSP1 and NSP13 significantly reduced NLRP3 inflammasome-induced caspase-1 activity by 46% and 39%, respectively. In addition, both NSP1 and NSP13 reduced mature IL-1β secretion by 31% and 28%, respectively (Figure 4B). Because the transfection efficiency was about 20%, non-transfected cells were stimulated with LPS and ATP, possibly increasing background caspase-1 activity. Neither NSP1 QM nor NSP13 ∆1A/2A had any effects on NLRP3 inflammasome-induced caspase-1 activity and mature IL-1β secretion. 

The effects of NSP1 and NSP13 on protein levels of components of the NLRP3 inflammasome were evaluated by Western blotting (Figure 4C,D). Treatment of THP-1 cells with LPS and ATP increased the levels of pro-IL-1β protein and induced the cleavage of caspase-1 into its active p20 subunit. Both NSP1 and NSP13 blocked the NLRP3-inflammasome-induced generation of caspase-1 p20. NSP1, but not NSP13, slightly reduced the levels of NLRP3, ASC and caspase-1 proteins. Neither NSP1 QM nor NSP13 ∆1A/2A had any effects on NLRP3-inflammasome-induced generation of caspase-1 p20 and the levels of NLRP3, ASC and caspase-1. Taken together, these results indicate that both NSP1 and NSP13 inhibit NLRP3 inflammasome activation by inhibiting caspase-1 activity in THP-1 cells. 

## 3. Discussion

Homeostatic regulation of inflammasome activation is critical for viral pathogenesis. Proper activation of inflammasomes promotes immune responses to eliminate infecting viruses. However, dysregulated activation of inflammasomes can be detrimental to the host by inducing immunopathogenesis [35].

SARS-CoV activates the NLRP3 inflammasome, with the expression of viral proteins translated from subgenomic RNAs, such as E, ORF3a and ORF8b, being responsible for NLRP3 inflammasome activation [21,22,23,24]. These viral proteins may directly activate the NLRP3 inflammasome, whereas a high viral load of SARS-CoV may activate the NLRP3 inflammasome indirectly. Hyper-induction of proinflammatory cytokines has been associated with the progression of diseases induced by infections with SARS-CoV and MERS-CoV [26,36,37,38]. MERS-CoV activation of the NLRP3 inflammasome is attenuated in the primary immune cells of bats, which act as asymptomatic viral reservoirs [39]. These findings suggest that the activation of NLRP3 inflammasomes by SARS-CoV proteins may contribute to the immunopathogenesis of viral diseases. 

SARS-CoV must also evade inflammasome responses and undergo primary replication at the site of infection prior to viral dissemination and immunopathogenesis. Our data indicate that SARS-CoV-2 NSP1 and NSP13, which are translated from genomic RNA, significantly inhibit the NLRP3 inflammasome. Because the amino acid residues of NSP1, which are critical for promoting RNA cleavage and suppressing host translation, are also required for caspase-1 inhibition, NSP1 likely interferes with the NLRP3 inflammasome complex by down-regulating the expression of its components, including NLRP3, ASC, caspase-1 and pro-IL-1β.

NSP13 also inhibits the NLRP3 inflammasome-induced cleavage of caspase-1 into its active p20 subunit without affecting host translation. NSP13 exhibits RNA helicase and nucleoside triphosphatase (NTPase) activities [40], with its helicase activity being dependent on its N-terminal ZBD [41]. Because both the N-terminal ZBD and the C-terminal helicase domain are essential for caspase-1 inhibition, NSP13 helicase activity may play an important role in inhibiting the NLRP3 inflammasome. Further studies are needed to determine the mechanism by which NSP13 inhibits caspase-1, as well as the roles of NSP1 and NSP13 in regulating the NLRP3 inflammasome. 

Inhibition of NLRP3 activation can be either protective or detrimental depending on the stage of viral infection [42]. Treatment of mice with the specific NLRP3 inhibitor, MCC90, within 24 h of infection with the lethal influenza A virus (IAV), significantly increased mortality [42]. However, inhibition of the NLRP3 inflammasome at later stages of IAV infection after disease onset significantly delayed viral pathogenesis [42].

Interestingly, both NSP1 and NSP13, which are translated during the early stage of SARS-CoV-2 infection, have been found to inhibit NLRP3 inflammasome activation, whereas E, ORF3a and ORF8b, which are translated during a later stage, promote NLRP3 inflammasome activation. SARS-CoV-2 may utilize NSP1 and NSP13 to evade inflammasome responses, allowing replication of its genome and production of structural and accessory proteins. Substantial amounts of viral proteins and nucleic acids produced in infected cells may activate the NLRP3 inflammasome, facilitating viral dissemination and pathogenesis. Further studies are needed to determine the mechanism by which SARS-CoV-2 regulates NLRP3 inflammasome activation to favor its life cycle.

## 4. Materials and Methods

### 4.1. Cells, Reagents and Transfection

The maintenance and propagation of HEK293T cells have been previously described [43]. Human monocytic THP-1 cells, purchased from the Korean Cell Line Bank (KCLB, Seoul, Korea), were cultured in RPMI 1640 (Hyclone, Logan, UT, USA) containing 10% fetal bovine serum, 1% penicillin/streptomycin and 0.05mM 2-mercaptoethanol. THP-1 cells were differentiated by incubation with 100 ng/mL PMA (Sigma-Aldrich, St. Louis, MO, USA) for 15 h, followed by replacement with fresh medium without PMA. The cells were incubated with 500 ng/mL LPS (L5293, Sigma-Aldrich) and 5 mM ATP (A2383, Sigma-Aldrich) to active NLRP3 inflammasomes. Cell supernatants were concentrated 30-fold using Amicon Ultra Centrifugal Filters 0.5 mL 10K (Millipore, Burlington, MA, USA). Transient transfection using Omicsfect™ was performed according to the manufacturer’s instructions (Omics Bio, Taipei city, Taiwan).

### 4.2. SARS-CoV-2 cDNA Library

pDONR223 SARS-CoV-2 ORFs and ORFs no-stop and pDONR207 SARS-CoV-2 ORFs and ORFs no-stop were gifts from Fritz Roth (Addgene, Watertown, MA, USA) [44]. Twenty-eight SARS-CoV-2 ORFs and ORFs no-stop were cloned into the vectors pEF1α-DEST and pEF1α-DEST with a C-terminal c-Myc-tag, respectively, from pENTR vector using LR clonase™ ll enzyme mix according to the manufacturer’s instructions (Invitrogen, Carlsbad, CA, USA). 

### 4.3. Plasmid Constructs

To generate pEF1α-DEST vector, EF1α- promoter was PCR amplified from pEBG, the kind gift of David Baltimore (Addgene plasmid #22227) [34] using the following primers: 5′-GGAAGAGCTCGCGAATGCATGTGCTCCGGTGCCCGTCAGTGGGCA-3′ (forward) and 5′- GCTGTTTCCTGTGTGAAATTGTTATCCGTTCACGACACCTGAAATGGAAGAAA-3′ (reverse). The PCR products were used as mega primers to PCR amplify pEF1α-DEST from pDEST-12.2. To generate pEF1α-DEST with a C-terminal c-Myc-tag, 6x-Myc in pCS3-MT was PCR amplified using the following primers: 5′- CAGCTTTCTTGTACAAAGTGGTGATCGCGGCAGGATCCCATCGATTTAAAGCT-3′ (forward) and 5′-TCACTATAGGGAGAGAGCTATGACGTCGGTGATCCTTGAATTCGAGATCTCTA AATT-3′ (reverse). The PCR products were used as mega primers to amplify pEF1α-DEST with a C-terminal c-Myc-tag from pEF1α-DEST.

To generate destination vectors expressing C-terminal c-Myc-tagged NSP1 and NSP13 mutants, pDONR223 SARS-CoV-2 NSP1 no-stop and pDONR223 SARS-CoV-2 NSP13 no-stop were PCR-amplified using an EZchange™ Site-directed Mutagenesis kit (Enzynomics, Daejeon, Korea). The primers used to generate NSP1 and NSP13 mutants were listed in Appendix A. 

To generate destination vectors expressing caspase-1 and IL-1β, cDNAs from THP-1 were PCR amplified. The primers for caspase-1 were 5′-GGGGACAAGTTTGTACAAAAAAGCAGGCTCCATGGCCGACAAGGTCCTGAAG-3′ (forward) and 5′- GGGGACCACTTTGTACAAGAAAGCTGGGTCATGTCCTGGGAAGAGGTAG-3′ (reverse); and the primers for IL-1β were 5′-GGGGACAAGTTTGTACAAAAAAGCAGGCTCCATGGCAGAAGTACCTGAGCT-3′ (forward) and 5′-GGGGACCACTTTGTACAAGAAAGCTGGGTCTTAGGAAGACACAAATTGCA-3′ (reverse). The PCR product was cloned into the vector pDONR using BP clonase™ ll enzyme mix (Invitrogen), and the inserted sequence in the pENTR vector was cloned into the vector pEF1α-DEST using LR clonase™ ll enzyme mix (Invitrogen).

### 4.4. Western Blot Analysis

Cells were harvested, and culture supernatants were concentrated 30-fold using Amicon filter. Equal amounts of proteins were electrophoresed on Mini-PROTEAN^®^ TGX™ gels (Bio-Rad Laboratories, Hercules, CA, USA) and transferred to nitrocellulose membranes as described [45]. Antibodies to NLRP3 (D2P5E), caspase-1 (D7F10) and IL-1β (3A6) were purchased from Cell Signaling Technology; antibodies to ASC (F-9) and c-Myc (9E10) were purchased from Santa Cruz Biotechnology; an anti-Tubulin (T5168) antibody was purchased from Sigma-Aldrich. Secondary peroxidase-labeled anti-mouse or anti-rabbit immunoglobulin G antibodies were purchased from Jackson ImmunoResearch Laboratories, West Grove, PA, USA.

### 4.5. Electroporation

THP-1 cells in wells of six-well plates were transfected using the Neon™ transfection system with 100 μL tips according to the manufacturer’s instructions (Invitrogen). Each well was filled with 2 mL RPMI Complete Growth Media without antibiotics and pre-incubated at 37 °C in a 5% CO_2_ incubator. The cells were centrifuged at 1000× *g* for 5 min, washed with PBS and resuspended at 2 × 10^7^ cells/mL in a 110 μL R buffer containing 11 μg of vectors. The electroporation settings were 1600 V, 10 ms and three pulses. After electroporation, 100 μL of transfected cells were transferred to each well of a six-well plate and incubated.

### 4.6. Enzyme-linked Immunosorbent Assay (ELISA)

Concentrations of human IL-1β in culture supernatant were measured using human IL-1β/IL-1F2 Quantikine ELISA kits according to the manufacturer’s instructions (DLB50; R&D systems, Minneapolis, MN, USA). Briefly, 8.0 × 10^5^ 293T cells in 2 mL DMEM complete medium were seeded in each well of a six-well plate. The supernatants of cells transfected with pEF1α-DEST expressing C-terminal c-Myc-tagged SARS-CoV-2 ORFs were harvested, and 30 μL of each supernatant were diluted 1:10 with 270 μL DMEM complete medium. A 200 μL aliquot of each diluted supernatant was used for ELISA. To measure IL-1β concentrations in the supernatants of THP-1 cells, cells were suspended at 2 × 10^7^ cells/mL in a 110 μL R buffer, and 100 μL of cells electroporated with DNA were added to each well of a six-well plate, along with 2 mL RPMI complete medium (without antibiotics). A 200μL aliquot of each supernatant was used for ELISA. Concentrations of mature human IL-1β were calculated using a standard curve. 

### 4.7. Caspase Glo^®^1 inflammasome Assay

Caspase-1 activities in culture supernatants were measured using Caspase Glo^®^1 inflammasome assay according to the manufacturer’s instructions (Promega, Madison, WI, USA). THP-1 cells were resuspended at 2 × 10^7^ cells/mL in a 110 μL R buffer, and 100 μL of cells electroporated with DNA were added to each well of a six-well plate along with 2 mL RPMI complete medium (without antibiotics). The following day, the electroporated THP-1 cells were differentiated using PMA for 15 h, fresh medium was added without PMA, and the cells were incubated overnight. The cells were treated with LPS for 15 h and ATP for 1 h. Two 50 μL aliquots of each concentrated supernatant were transferred to wells of a 96-well plate; 50 μL of Caspase Glo^®^1 reagent was added to one aliquot and 50 μL of Caspase Glo^®^1 YVAD-CHO reagent was added to the other, followed by mixing on a plate shaker at 300 rpm for 30 s. After 2 h at room temperature, the luminescence in each well was measured using a Glomax Multi Detection System (Promega), with the luminescence in culture medium without cells subtracted from the luminescence measured in wells containing cells.

### 4.8. Quantitative Reverse Transcription PCR (qRT-PCR)

Total RNA was isolated from cells using HiGene^TM^ Total RNA Prep kits according to the manufacturer’s instructions (Biofact), followed by reverse transcription to cDNA using TOPscript™ cDNA Synthesis kits as described (Enzynomics). To analyze transcript levels, qRT-PCR was performed with the following primers: caspase-1, 5′- TTTCTGCTCTTCCACACCAG-3′ (forward) and 5′-CTCCACATCACAGGAACAGG -3′ (reverse); IL-1β, 5′-ACAGATGAAGTGCTCCTTCCA-3′ (forward) and 5′- GTCGGAGATTCGTAGCTGGAT-3′ (reverse); GAPDH. 5′-CATGAGAAGTATGACAACAGCCT-3′ (forward) and 5′-AGTCCTTCCACGATACCAAAGT-3′ (reverse). GAPDH was used as a reference gene to normalize the mRNA expression.

## Figures and Tables

**Figure 1 microorganisms-09-00494-f001:**
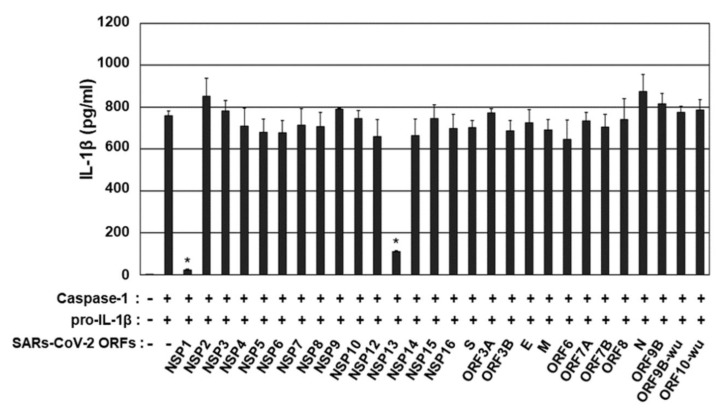
Screening of severe acute respiratory syndrome coronavirus 2 (SARS-CoV-2) open reading frames (ORFs) that regulate caspase-1-mediated IL-1β production. Human embryonic kidney 293T (HEK 293T) cells were co-transfected with vectors expressing caspase-1 and pro-IL-1β, plus either pEF1α-DEST or pEF1α-DEST expressing complementary DNAs (cDNAs) encoding 28 SARS-CoV-2 ORFs. After 24 h, mature IL-1β in culture supernatants was measured by enzyme-linked immunosorbent assay (ELISA). * *p* < 0.001 by unpaired two-tailed Student’s *t*-tests. Data shown are representative of three independent experiments.

**Figure 2 microorganisms-09-00494-f002:**
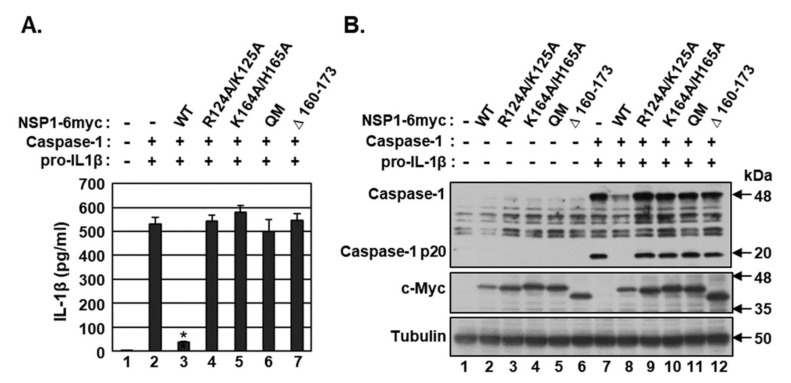
Determination of the amino acid residues of NSP1 critical for caspase-1 inhibition. (A and B) HEK 293T cells were co-transfected with vectors expressing caspase-1 and pro-IL-1β plus either pEF1α-DEST or pEF1α-DEST expressing c-Myc-tagged NSP1 wild type (WT) or NSP1 containing the mutations R124A/K125A, K164A/H165A, and both R124A/K125A and K164A/H165A (quadruple mutant (QM)) and a deletion of amino acids 160–173 (∆ 160–173). After 24 h, (**A**) the concentrations of mature IL-1β secreted in culture supernatants were measured by ELISA, and (**B**) levels of caspase-1 and NSP1 (c-Myc) proteins and the cleavage of caspase-1 into its p20 subunit were determined by Western blotting. ELISA data shown are representative of three independent experiments. * *p* < 0.001 by unpaired two-tailed Student’s *t*-tests.

**Figure 3 microorganisms-09-00494-f003:**
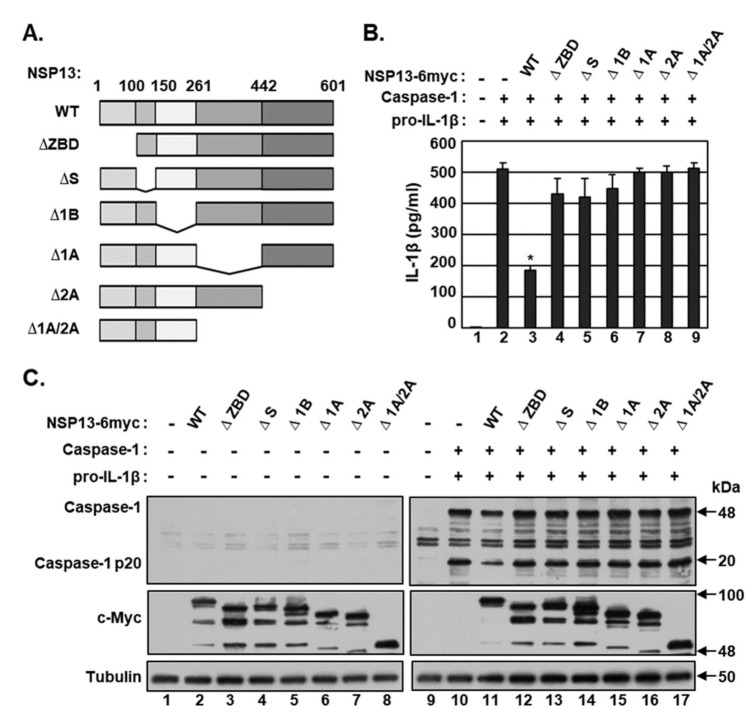
Determination of the NSP13 domain required for caspase-1 inhibition. (**A**) Diagrammatic representation of NSP13 deletion mutants. (B and C) HEK 293T cells were co-transfected with vectors expressing caspase-1 and pro-IL-1β plus either pEF1α-DEST or pEF1α-DEST expressing c-Myc-tagged NSP13 WT or mutants carrying deletions of zinc-binding domains (ZBD) (∆ZBD), the stalk domain (∆S), the 1B domain (∆1B), the two RecA-like domains, 1A (∆1A) and 2A (∆2A) and both 1A and 2A (∆1A/2A). After 24 h, (**B**) the concentrations of mature IL-1β secreted in culture supernatants were measured by ELISA, and (**C**) levels of caspase-1 and NSP13 (c-Myc) proteins and the cleavage of caspase-1 into its p20 subunit were determined by Western blotting. ELISA data are representative of three independent experiments. * *p* < 0.001 by unpaired two-tailed Student’s *t*-tests.

**Figure 4 microorganisms-09-00494-f004:**
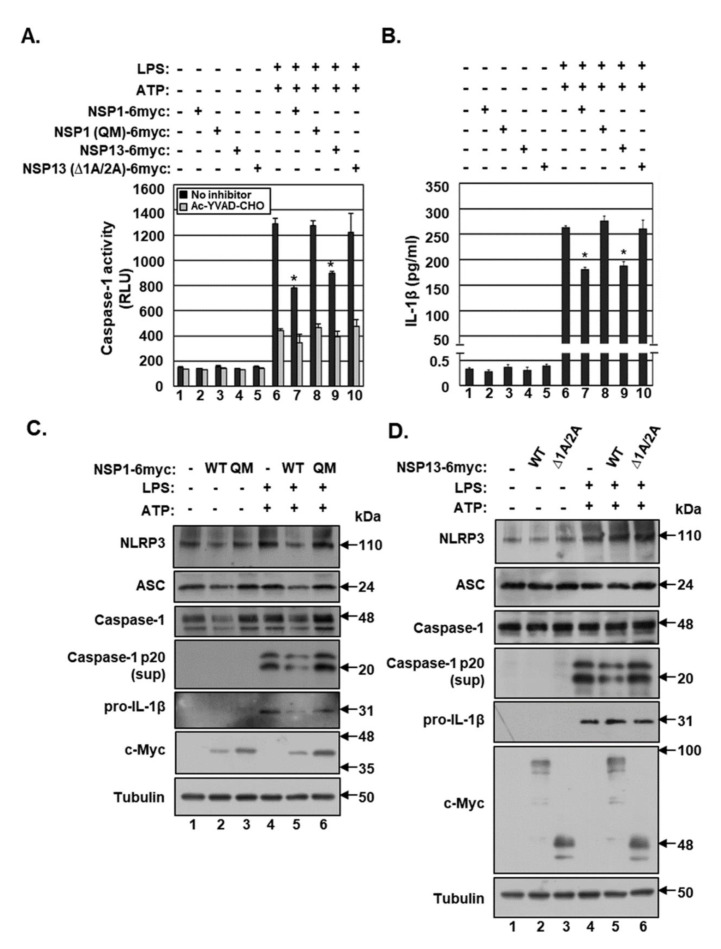
Effects of NSP1 and NSP13 on the NLRP3 inflammasome in THP-1 cells. THP-1 cells were electroporated with either pEF1α-DEST or pEF1α-DEST expressing c-Myc-tagged NSP1 WT, NSP1 QM, NSP13 WT or NSP13 ∆1A/2A. Twenty hours later, the cells were differentiated by incubation with phorbol-12-myristate 13-acetate (PMA), followed by treatment with lipopolysaccharide (LPS) for 15 h and adenosine triphosphate (ATP) for 1 h. (**A**) Caspase-1 activity measured by Caspase Glo^®^1 Inflammasome assays. (**B**) Mature IL-1β secretion in culture supernatants measured by ELISA. (C and D) Protein levels of NLRP3, ASC, caspase-1, caspase-1 p20, IL-1β, c-Myc and tubulin in THP-1 cells expressing (**C**) NSP1 WT and QM and (**D**) NSP13 WT and ∆1A/2A determined by Western blotting. The results of Caspase Glo^®^1 inflammasome assays and IL-1β ELISA are representative of three independent experiments. * *p* < 0.001 by unpaired two-tailed Student’s *t*-tests. Sup, supernatant.

## Data Availability

The data presented in this study are available on request from the corresponding author.

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
