# Peer review of "SARS-CoV-2 Nonstructural Proteins 1 and 13 Suppress Caspase-1 and the NLRP3 Inflammasome Activation"

_microorganisms, 2021, doi:10.3390/microorganisms9030494_

Round 1

Reviewer 1 Report

The manuscript is well documented.

  1. The title needs to be rephrased.
  2. Are we talking about dysregulated activation which means activation is present but in deformity or suppression or activation which implies no activation. Please use the technical terms stringently.
  3. Lines 14 & 15 sound confusing please rephrase. I would advice to structure the abstract first with background information of healthy individuals and then adding information about Sars infected ones, to highlight the contrast which is the major theme of your manuscript.
  4. Abstract needs Grammar modifications.
  5. There is a format error in line #36.
  6. Line # 98 needs rephrasing to reflect cause and then affect.
  7. Line # 128 needs to be checked for word choice error.
  8. Line # 194 grammar correction is needed.
  9. Primer information can be better represented in tabular form.

Author Response

To: Ms. Natalija Verovic, MSc, Assistant Editor, MDPI

Dear Ms. Verovic,

Please find below our response to the reviewers for the manuscript #microorganisms-1109929 (Title: SARS-CoV-2 nonstructural proteins 1 and 13 suppress caspase 1 and activation of the NLRP3 inflammasome) item by item. We highly appreciate the reviewers’ comments and your assistance. 

Best regards,

Yoon-Jae Song, Ph.D.

Reviewer #1: 

The manuscript is well documented.

Point 1: The title needs to be rephrased.

Response 1: As suggested by the reviewer, the title has been changed to “SARS-CoV-2 nonstructural proteins 1 and 13 suppress caspase-1 and the NLRP3 inflammasome activation”.

Point 2: Are we talking about dysregulated activation which means activation is present but in deformity or suppression or activation which implies no activation. Please use the technical terms stringently.

Response 2: As described in the previous report (Schroder, K. and J. Tschopp. 2010. The Inflammasomes. Cell. 140:821-832), dysregulated inflammasome activity has been implicated in the pathogenesis of inflammatory and infectious diseases. Both hyper- and hypo-activation of inflammasomes against viral infections can be deleterious to the host. As described in other reports, dysregulated inflammasome activity usually implies an uncontrolled-hyperactivation of inflammasome.

Point 3: Lines 14 & 15 sound confusing please rephrase. I would advice to structure the abstract first with background information of healthy individuals and then adding information about Sars infected ones, to highlight the contrast which is the major theme of your manuscript.

Response 3: As described in the abstract, proper inflammasome activation protects the host from viral infections. However, dysregulated inflammasome activation can cause unwanted, immune pathogenesis which can be detrimental to the host. SARS-CoVs have been reported to induce hyperactive inflammatory responses to cause immune pathogenesis. However, SARS-CoVs must also evade inflammatory responses to establish primary replication before they spread and cause diseases in the host. The abstract (lines 13-17) has been revised as follows: “Coronaviruses including SARS-CoV and MERS-CoV encode viroporins that activate the NLRP3 inflammasome, and the severity of coronavirus disease is associated with the inflammasome activation. Although the NLRP3 inflammasome activation is implicated in the pathogenesis of coronaviruses, these viruses must evade inflammasome-mediated antiviral immune responses to establish primary replication.”

Point 4: Abstract needs Grammar modifications.

Response 4: Grammar errors have been reviewed. The manuscript was revised with a professional English editing service.

Point 5: There is a format error in line #36.

Response 5: The space has been removed.

Point 6: Line # 98 needs rephrasing to reflect cause and then affect.

Response 6: Line #98 implies that the amino acid residues of NSP1, which are critical for host translation shutoff, are also required for caspase-1 inhibition.

Point 7: Line # 128 needs to be checked for word choice error.

Response 7: Line #128 implies that NSP13 protein requires both the N- and C-termini to inhibit caspase-1.

Point 8: Line # 194 grammar correction is needed.

Response 8: Grammar errors have been reviewed.

Point 9: Primer information can be better represented in tabular form.

Response 9: As suggested, a supplementary table 1 has been included in the manuscript to provide the list of primers to generate NSP1 and NSP13 mutants.

Reviewer 2 Report

In this study, the authors screened a cDNA library of 28 SARS-CoV-2 ORFs and identified two nonstructural proteins (NSPs), NSP1 and NSP13, inhibiting NLRP3 inflammasome-induced caspase-1 activity and IL-1β secretion. Since a dysregulation of the NLRP3 inflammasome was shown to be associated with the severity of coronavirus infection, this work provides new insights into the mechanisms underlying SARS-CoV-2 pathogenesis. The manuscript is well written and It is easy to follow the authors’ argumentation since the presented results support their conclusions nicely. However, one big drawback is clearly the absence of functional data in the context of virus infection. Therefore, the relevance of their findings in an infection context is not really clear and can only be speculated on. Some additional comments are given below.

How did the authors control for transfection efficiencies and ORF expression in their screen? Several other studies, both with SARS-CoV-1 as well as preliminary data with SARS-CoV-2, have shown an activation of NLRP3 by ORF3a as well as E, however in their screen the authors do not see an increase in IL-1b after overexpression. How do they explain these discrepancies?

Minor comments:

Please provide the size of the western blot bands eg. upon inclusion of a ladder.

Line 36: S, E, M, and N, along (missing point)

Line 90: IL-1β secretion by 91% and 71%

Author Response

To: Ms. Natalija Verovic, MSc, Assistant Editor, MDPI

Dear Ms. Verovic,

Please find below our response to the reviewers for the manuscript #microorganisms-1109929 (Title: SARS-CoV-2 nonstructural proteins 1 and 13 suppress caspase 1 and activation of the NLRP3 inflammasome) item by item. We highly appreciate the reviewers’ comments and your assistance. 

Best regards,

Yoon-Jae Song, Ph.D.

Reviewer #2: 

In this study, the authors screened a cDNA library of 28 SARS-CoV-2 ORFs and identified two nonstructural proteins (NSPs), NSP1 and NSP13, inhibiting NLRP3 inflammasome-induced caspase-1 activity and IL-1β secretion. Since a dysregulation of the NLRP3 inflammasome was shown to be associated with the severity of coronavirus infection, this work provides new insights into the mechanisms underlying SARS-CoV-2 pathogenesis. The manuscript is well written and It is easy to follow the authors’ argumentation since the presented results support their conclusions nicely. However, one big drawback is clearly the absence of functional data in the context of virus infection. Therefore, the relevance of their findings in an infection context is not really clear and can only be speculated on. Some additional comments are given below.

Point 1: How did the authors control for transfection efficiencies and ORF expression in their screen? Several other studies, both with SARS-CoV-1 as well as preliminary data with SARS-CoV-2, have shown an activation of NLRP3 by ORF3a as well as E, however in their screen the authors do not see an increase in IL-1b after overexpression. How do they explain these discrepancies?

Response 1: Transfection efficiencies and expression of SARS-CoV-2 ORFs that were used for the screening were confirmed by qRT-PCR. For the screening, HEK293T cells were transfected with vectors expressing caspase-1 and pro-IL-1β plus the expression vectors for SARS-CoV-2 ORFs, and effects of SARS-CoV-2 ORFs on caspase-1-mediated IL-1b activation was determined by ELISA. ORF3a and E are viroporins and mediate potassium and calcium ion efflux to activate the NLRP3 inflammasome. Since the components of the NLRP3 inflammasome that are upstream of caspase-1 are absent in HEK293T cells, we expected that ORF3a and E have no effect on IL-1b activation. On the other hand, we found that mature IL-1b secretion in culture supernatants of THP-1 cells electroporated with expression vectors for SARS-CoV-2 ORF3a and E (data not shown). As previously reported, ORF3a and E activate the NLRP3 inflammasome in THP-1, but not in HEK293T, cells.

Minor comments:

Point 2: Please provide the size of the western blot bands eg. upon inclusion of a ladder.

Response 2: As suggested, the size of the bands has been included in figures 2, 3 and 4.

Point 3: Line 36: S, E, M, and N, along (missing point)

Response 3: The space has been removed.

Point 4: Line 90: IL-1β secretion by 91% and 71%

Response 4: A change has been made as suggested.